# Incidence, Risk Factors, and Prevention Strategies for Post-ERCP Pancreatitis in Patients with Biliopancreatic Disorders and Acute Cholangitis: A Study from a Romanian Tertiary Hospital [note 1]

**DOI:** 10.3390/biomedicines13030727

**Published:** 2025-03-17

**Authors:** Matei-Alexandru Cozma, Cristina Angelescu, Andrei Haidar, Radu Bogdan Mateescu, Camelia Cristina Diaconu

**Affiliations:** 1Faculty of Medicine, “Carol Davila” University of Medicine and Pharmacy, 050474 Bucharest, Romania; 2Department of Gastroenterology and Hepatology, Colentina Clinical Hospital, 020125 Bucharest, Romania; 3Department of Internal Medicine, Clinical Emergency Hospital of Bucharest, 105402 Bucharest, Romania

**Keywords:** post-ERCP pancreatitis, main pancreatic duct cannulation, endoscopic papillary balloon dilatation

## Abstract

**Background/Objectives**: Pancreatitis is the most frequent and serious complication of endoscopic retrograde cholangiopancreatography (ERCP), with an incidence between 2 and 10% and a mortality rate of 1 in 500 patients. Etiopathogenesis remains poorly understood. The aim of this study was to analyze the incidence of post-ERCP pancreatitis (PEP) and to identify potential patient- and procedure-related risk factors (RF) in a cohort of patients from a tertiary referral center in Romania. **Methods**: We conducted a retrospective, observational, single-center study in which we analyzed ERCP procedures performed in the Gastroenterology Department of Colentina Clinical Hospital, Bucharest, Romania, between January 2019 and September 2024. All patients received intrarectal diclofenac before the ERCP and were hydrated with at least 1500 mL of Ringer’s solution after the procedure in the absence of contraindications, according to the latest international recommendations. **Results**: In total, 2743 ERCPs were performed in the given time period, while 2350 procedures were analyzed in the study. PEP occurred in 350 cases (14.9%). Of these, 191 (54.6%) occurred in males with a mean age of 66.5 years. Procedural RF with adjusted odds ratios (OR) were as follows: difficult cannulation of the common bile duct, OR = 3.734, *p* < 0.001, main pancreatic duct catheterization, OR = 1.454, *p* = 0.022, and endoscopic papillary balloon dilatation, with an OR of 3.258, *p* < 0.001. Pancreatic duct stent placement was shown to prevent PEP in this study group (*p* < 0.001). **Conclusions**: PEP remains a serious complication of ERCP, associated with significant morbidity and occasional mortality. While some proven risk factors, such as age, gender, or comorbidities, are unmodifiable, avoiding Wirsung duct cannulation and pancreatography, or prophylactic pancreatic duct stent placement, could play a significant role in PEP prevention.

## 1. Introduction

Endoscopic retrograde cholangiopancreatography (ERCP) is the most commonly used endoscopic procedure for the treatment of biliopancreatic tract disorders. Despite it’s widespread adoption in numerous endoscopy units globally and the improvement in the learning curve through the introduction of multiple training centers for young endoscopists, ERCP remains the endoscopic procedure associated with the highest complication rate, estimated between 0.5 and 7% and mortality up to 0.2% [1,2,3,4]. In addition to post-sphincterotomy bleeding, cholangitis, cholecystitis, and perforation, post-ERCP pancreatitis (PEP) represents the most common and serious ERCP-related adverse event, with an incidence generally ranging between 5 and 10% [5,6]. However, incidence can be as high as 40% in certain high-risk populations or when ERCP is performed outside expert, high-volume centers [2,5,6]. Most cases of PEP are mild, while severe forms occur only in 0.3 to 0.8% and are usually associated with complications including multiple organ failure, pancreatic necrosis, peripancreatic collections, and high related health costs [5,7].

The etiopathogenesis is likely related to a combination of factors, such as chemical or allergic injury from contrast agent injection, as well as mechanical obstruction or increased pressure within the main pancreatic duct (MPD). This usually results from periampullary inflammation and edema caused by normal or over-manipulation of the papilla during ERCP, ultimately activating the pancreatic enzymes [3,8]. The most frequent procedure-related risk factors (RF) for PEP are difficult cannulation of the common bile duct (CBD), repeated guidewire (GW) cannulation or contrast material injection into the MPD, and endoscopic papillary balloon dilatation (EPBD) [1,2,3]. Precut sphincterotomy has also been associated with an increased risk for PEP when it is performed after repeated cannulation attempts using other methods [3,9]. Many research studies also identified multiple patient-related RF such as female sex, personal history of acute pancreatitis (AP), age < 55 years old, or normal total bilirubin (TB) levels [8,10,11]. Despite numerous variations, the most recently accepted definition was introduced by Cotton and colleagues in 1991 and revised in 2012 by the Atlanta Consensus. The revised definition requires a new or increased abdominal pain consistent with AP, a serum lipase level of more than three times the upper limit of normal (ULN) within 24 h of the procedure or the need for new or prolonged hospitalization [7,12,13].

Despite the absence of a proven effective method to limit the severity of PEP, there have been many attempts to prevent and minimize the risk of PEP [10]. Prophylaxis involves a combination of patient risk stratification and pharmacological or procedural measures. Identifying patients at high risk of PEP helps tailor preventive measures such as careful patient selection to avoid unnecessary exposure to ERCP and its related risks. It has already been known that rectal administration of 100 mg of indomethacin or diclofenac immediately before or after ERCP, as well as aggressive intravenous hydration in certain high-risk patients, help reduce the risk of developing PEP [14]. Regarding procedural measures, minimizing the number of attempts needed for CBD cannulation and MPD stent placement has been proven to effectively reduce the risk of PEP [3,14].

This study aimed to identify and evaluate potential patient- and procedure-related RF associated with PEP development in a cohort of patients treated at a tertiary referral hospital in Bucharest, Romania. Additionally, the study aimed to compare the results obtained with existing data in the literature and to provide a comprehensive understanding of these RF and their role in the etiopathogenesis and evolution of PEP. The ultimate goal was to enhance preventive strategies and optimize clinical outcomes, contributing to better patient care and more effective procedural protocols.

## 2. Materials and Methods

### 2.1. Study Design and Patient Selection

To extensively characterize the episodes of PEP and to identify potential RF associated with its etiopathogenesis, we conducted a retrospective case-control study in which we included all patients who underwent an ERCP, regardless of the indication or endoscopic history of the patients, between January 2019 and September 2024 within the Gastroenterology and Hepatology Department of Colentina Clinical Hospital, a tertiary referral hospital in Bucharest, Romania. This endoscopy unit also serves as an ERCP training center for fellows in the field, providing a robust and diverse patient population for analysis. All ERCP procedures were performed by or under close supervision of senior endoscopists in collaboration with dedicated endoscopy nursing staff. Procedures were carried out under deep sedation, with continuous monitoring by an anesthesiologist to ensure patient safety and procedural efficiency. Post-procedural monitoring was conducted in an intensive care setting for a minimum of 2–4 h, allowing for the early detection and management of complications, including PEP.

Diagnosis of PEP was established following the latest recommendations of the European Society of Gastrointestinal Endoscopy (ESGE), which define PEP as a condition characterized by new or worsening abdominal pain accompanied by elevated pancreatic enzyme levels (amylase or lipase more than three times the ULN) within the first 24 h after an ERCP [15]. Serum lipase measurements were obtained only in cases where clinical symptoms were suggestive of AP. Cross-sectional imaging techniques, such as computed tomography (CT) and magnetic resonance imaging (MRI), were not routinely performed and were reserved for severe cases. The severity of PEP was further categorized using the revised Atlanta classification criteria published in 2012 [16].

Prophylaxis for PEP was implemented according to the latest ESGE and American Society for Gastrointestinal Endoscopy (ASGE) guidelines. Preventive measures included the rectal administration of 100 mg of diclofenac immediately before or after ERCP and aggressive post-procedural intravenous hydration using a 20 mL/kg bolus of Ringer’s solution, followed by a continuous infusion at 3 mL/kg/h. These strategies aimed to mitigate the risk of pancreatitis and improve patient outcomes. The Ethics Committee of Colentina Clinical Hospital approved this study under registration number 10/12.09.2022.

### 2.2. Data Acquisition and Study Variables

Data for this study were collected from multiple sources, including the electronic hospital database, patient medical charts, discharge summaries, imaging records, and laboratory files. Variables were analyzed to ensure a detailed assessment of both patient and procedure-related factors. These included demographics (age and sex), clinical history (comorbidities, prior endoscopic interventions or episodes of AP), symptoms (clinical presentations recorded at admission and after ERCP), laboratory data (serum lipase and TB levels), procedure-related variables (degree of difficulty in cannulating the CBD, the number of unintentional MPD cannulations or contrast agent injection, the use of precut sphincterotomy, placement of prophylactic pancreatic stents, endoscopic papillary balloon dilatation—EPBD) and outcomes (prolongation of hospital stay, the incidence of PEP, and other procedural complications). This dataset enabled a thorough evaluation of the potential RF and their associations with PEP incidence and severity.

### 2.3. Statistical Analysis

Statistical analyses were conducted using SPSS software version 29.0 (IBM, Endicott, NY, USA) and GraphPad Prism v9.2.0 (GraphPad, San Diego, CA, USA) for graphical representation of the data. Continuous variables were expressed as mean values with standard deviations (SD), while categorical variables were reported as frequencies (n) and percentages (%). Comparative analyses between study groups were performed using the Chi-square test or Fisher’s exact test, depending on the data distribution. A *p*-value of <0.05 was considered statistically significant.

Binary logistic regression analysis was employed to identify causal relationships between potential RF and the incidence of PEP. This approach allowed for the evaluation of the impact of certain RF proposed by the ESGE and ASGE guidelines on the likelihood of PEP occurrence and its subsequent clinical evolution in our study cohort. The logistic regression model provided adjusted odds ratios (OR) with 95% confidence intervals (CI), facilitating an accurate assessment of the independent contribution of each variable to the risk of PEP. This multi-faceted approach to statistical analysis ensured robust and reliable results, supporting evidence-based conclusions regarding the risk factors and prevention strategies for PEP.

## 3. Results

The present study aimed to analyze a cohort of patients who underwent an ERCP procedure over a five-year period at Colentina Clinical Hospital. The main objective was to identify potential RF related to patient or endoscopic techniques that could lead to PEP development.

Overall, within the Gastroenterology and Hepatology Department of Colentina Clinical Hospital, 2743 ERCP procedures were performed between January 2019 and September 2024, regardless of indication. We excluded patients with a lack of follow-up after ERCP (patients who were transferred to other medical units immediately after the procedure) or incomplete data, especially those with incomplete endoscopic reports, with missing data such as the degree of difficulty in cannulating the CBD, precut sphincterotomy or pancreatic stenting placement), and patients who developed other complications not related to ERCP. After applying the exclusion criteria, a total of 2350 procedures were eligible for the study. Of these, 350 cases (14.9%) developed PEP. The patient’s inclusion process is illustrated in Figure 1.

The mean age was comparable between the two groups, with 66.5 years old for the PEP group and 66.25 for the non-PEP group. Gender distribution showed a significantly higher proportion of males in the PEP group (191, 54.6%) compared to the non-PEP group (952, 47.6%), with a *p*-value of 0.018, but with a female predominance in the total study group with 52.3%. Comorbidities such as obesity and type 2 diabetes mellitus (T2DM) were more frequent in the PEP group, with T2DM affecting 91 patients (26%) compared to 391 patients (19.5%) in the non-PEP group (*p* = 0.007) and obesity 47 patients (13.4%) compared to 227 (11.4%) (non-significant difference).

In terms of personal medical history, in the PEP group, 19 patients (5.4%) had a previous episode of AP compared to 97 (4.9%) in the control group, and 58 (16.6%) had a history of previous biliary instrumentation, compared to 309 (15.45%) in the non-PEP group. Mortality was significantly higher in the PEP group, eight patients (2.29%) compared to the non-PEP group, 17 (0.85%), *p* = 0.015, with an overall mortality rate of 1.05%. The study data regarding the general characteristics of the population are detailed in Table 1.

Regarding the PEP group, the vast majority of episodes—337 (96.3%)—were mild forms, while severe cases were represented by only four episodes (1.1%). The mean serum lipase level in this group was 1955.44 U/L (Table 2).

CT-scanning was performed in only two patients, and imaging features included in both cases diffuse enlargement and heterogeneous attenuation of the pancreas, peripancreatic inflammation with reticular stranding of the surrounding fat, and small areas of pancreatic necrosis.

The most frequent indication for ERCP across the entire cohort was choledocholithiasis, accounting for 802 (34.1%) of all cases, 30 (8.5%) in the PEP group, and 772 (38.6%) in the control group. Other significant indications were represented by pancreatic head cancer, observed in similar proportions in the PEP and in the control group (24.9% and 24.4%, respectively), hilar cholangiocarcinoma, which was significantly more common in the PEP group (140 cases, 40%) compared to the non-PEP group (197 cases, 9.9%), *p* < 0.0001, indicating a potential procedural challenge and RF, distal cholangiocarcinoma 23 (6.6%) and 206 cases (10.3%), *p* = 0.03, ampullary tumors 19 (5.4%) and 113 cases (5.7%), and distal cholangiocarcinoma, with 23 (6.6%) and 206 (10.3%), *p* = 0.03. All indications for ERCP are detailed in Table 3 below and illustrated in Figure 2.

Procedural-related RF was found to be significantly associated with the development of PEP. Difficult cannulation of the CBD was observed in 50.3% of the cases (n = 175) in the PEP group, compared to 22% (n = 440) in the non-PEP group, *p* < 0.0001. Precut sphincterotomy was performed more frequently in the PEP group (21.1% vs. 15.6%, *p* = 0.0186), as well as main pancreatic duct catheterization and contrast agent injection, 36.6% vs. 26% (*p* < 0.0001) and 15.7% vs. 10.4% (*p* = 0.0043). Moreover, prophylactic placement of a pancreatic duct stent was less common in the PEP group (7.7% vs. 10.9%), thus suggesting a possible under-use of this tool.

The presence of a duodenal diverticulum was noted in 14% of PEP patients compared to 8.6% in the non-PEP group (*p* = 0.002). Other notable RF included endoscopic papillary balloon dilatation (EPBD), which was significantly more used in the PEP group (20.6% vs. 4.7%, *p* < 0.0001), and the presence of other complications during or after the procedure (4.3% vs. 2.5%). All the ERCP-related RF are detailed in Table 4 and illustrated in Figure 3.

In the next section, we focused on the postulated RF and whether they are correlated with PEP development in our study group. For this purpose, we performed a logistic regression using the occurrence of PEP as a dependent variable and the RF and the characteristics of the study group that were already mentioned as co-variables. The results are highlighted in detail in Table 5 and illustrated as a Forest plot chart in Figure 4. Thus, the factors associated with PEP development in patients who underwent ERCP in our study group were T2DM, with an odds ratio (OR) of 1.508 (1.069–2.128), *p* = 0.019, several indications for ERCP such as benign and malignant ampuloma, OR of 5.487 (1.94–15.516), *p* < 0.001, distal cholangiocarcinoma, OR of 3.255 (1.178–8.992), *p* < 0.001, and external compression of the CBD by liver or lymph node metastases, chronic pancreatitis, or Mirizzi syndrome, OR of 5.677 (1.173–27.487), *p* = 0.031, TB levels, OR = 1.049 (1.023–1.075), *p* < 0.001, difficult cannulation of the CBD, OR = 3.734 (2.747–5.076), *p* < 0.001, MPD catheterization, OR = 1.454 (1.056–2.002), *p* = 0.022, and EPBD, with an OR of 3.258 (2.011–5.28), *p* < 0.001. On the other hand, the only procedural variable analyzed that showed a significant negative association was the MPD stent placement, with an OR of 0.485 (0.27–0.873) and a *p*-value of 0.016, thus proving the role it may have in PEP prophylaxis in high-risk patients.

## 4. Discussion

The role of ERCP in the treatment of intra- or extrahepatic obstructions of the biliary tract remains central. At the same time, it is associated with the highest incidence of adverse reactions, being a demanding and complex endoscopic procedure even for experienced endoscopists. PEP remains the most common and serious complication encountered, and although some proven preventive measures are widely implemented, such as the administration of intrarectal nonsteroidal anti-inflammatory drugs (NSAIDs) and intravenous hydration, the incidence of PEP remains around 5–10%, and the mortality rate is around 0.2% [2,7]. In addition to patient-related, non-modifiable RF, procedure-related RF is considered to have the greatest influence on the etiopathogenesis of PEP.

Serum lipase levels are a key element in the diagnosis of PEP, usually performed after the onset of symptoms suggestive of AP. Transient asymptomatic elevation of serum lipase or amylase occurs in up to 70% of patients after ERCP, beginning at 90 min and peaking approximately 4 h after the procedure. Although this is well known in the endoscopic literature, numerous studies have shown that a value of more than three to four times the ULN in patients without suggestive symptoms of AP performed two to six hours after ERCP can help an early diagnosis of PEP, avoid underdiagnosis. At the same time, a value lower than three to four times the ULN has a sufficiently good negative predictive value for excluding PEP [17,18,19]. Because ERCP is nowadays more commonly performed in an outpatient setting, this practice is used to make discharge decisions after the procedure. For this reason, the latest ESGE guideline on PEP prevention recommends routine determination of serum lipase two to six hours after ERCP in patients presenting with pain suggestive of AP or in patients to be discharged on the day of ERCP [20].

In our center, despite few selected cases in which serum lipase level is measured within two to six hours after ERCP, usually in patients with a very difficult ERCP, in whom the MPD was extensively cannulated or injected inadvertently, or in patients in whom the indication for ERCP was pancreatic pathology (i.e., MPD lithiasis in the setting of chronic pancreatitis), in the vast majority of patients, serum lipase levels are determined only in the presence of symptoms suggestive to AP. This practice allows us to avoid an over-diagnosis of PEP, which would lead to over-hospitalization, supplementary imaging without any additional clinical benefit, or over-treating, which is not without risks. We also opted to perform cross-sectional imaging techniques just in selected cases, usually severe forms, in the assessment of complications, per international specialized guidelines [21].

The incidence of PEP is estimated, according to the latest literature data, between 5 and 10%. Bishay K. et al. have shown in their meta-analysis published in 2025 an approximately steady incidence since 2000, of 4.6% in all patients and 6.4% in patients without a history of previous sphincterotomy [2]. In our study, the incidence rate of PEP was 14.9%, considerably higher than those reported in the literature. Differences in incidence between different medical centers are described in the literature, and the reasons are multiple, both patient- and procedure-related. The most frequently cited are different categories of patients treated in certain centers (some referral centers predominantly treat patients with complex biliopancreatic disorders, many of them with a previous history of endoscopic sphincterotomy or episodes of AP), differences in the routine endoscopic techniques used or in the level of experience of the endoscopists who performed ERCP, and differences in the criteria for defining AP and the cut-off values of serum lipase levels sampled after ERCP and considered diagnostic. In our study, we consider that two main factors led to a higher incidence of PEP: the diverse pathology of the patients, a significant number of them being diagnosed with neoplastic biliopancreatic pathology and with a history of endoscopic sphincterotomy, and the different levels of expertise of the endoscopists who performed ERCP, our clinic being also a training center for young endoscopists.

Female gender and age < 55 years are considered important RF for developing PEP [2,3,4,5]. However, within the subgroup of patients who developed PEP, the mean age was 66.5 years old, and most patients were male, representing 54.6% of patients. However, multivariate analysis revealed that female sex was not a risk factor for PEP, with a *p*-value of 0.404.

The most important procedure-related RF are difficult cannulation of the CBD, repeated GW cannulation or injection of contrast agent into the MPD, EPBD, and precut sphincterotomy [8,9,10]. Numerous prospective randomized studies, as well as systematic reviews and meta-analyses on different technical variants of ERCP, have been published in recent years.

Primary CBD cannulation success is a main quality indicator of a successful ERCP procedure and, therefore, an area of interest in our study, being considered one of the main factors associated with the incidence of PEP. Traditionally, CBD cannulation has long been performed by the classic contrast-assisted cannulation (CAC) technique. However, this paradigm has shifted in recent years, currently in favor of the guide wire-assisted cannulation (GWAC) technique, which has shown superiority in CBD cannulation rate, decreased need for precut sphincterotomy, and overall, a lower incidence rate of PEP, as shown in multiple prospective studies and meta-analyses [1,6,22]. For example, in a prospective study conducted by Artifon et al., GWAC was associated with a significantly lower probability of PEP, with an OR of 0.43, a 95% CI of 0.21–0.89, and a *p*-value of 0.02 [23]. Moreover, a study published by Nambu T. et al. indicated similar data, showing an incidence of PEP of 3% (GWAC) vs. 6.5% (CAC) [24]. However, there are also some randomized studies, such as those published by Kawakami H. et al., Bailey A. et al., and Mariani A. et al., which demonstrated that, although the GWAC technique improves the primary success rate for CBD cannulation during ERCP, it was not associated with a decrease in the incidence of PEP [6,25,26]. Such a comparison was not possible in our study because all endoscopists who perform ERCP used the GWAC technique with sphincterotomes as the main method of CBD cannulation, while the CAC technique is not used anymore. However, not all the endoscopists in our clinic reported the number of CBD cannulation attempts performed during the procedure. Rather, it was noted as “easy” or “difficult” cannulation, associated or not with unintentional MPD cannulation. In our study, “difficult” CBD cannulation was strongly associated with an increased incidence of PEP, with an OR of 3.734, a 95% CI of 2.747–5.076, and a *p*-value < 0.001.

Precut sphincterotomy is usually used as a salvage method of CBD cannulation, often after prolonged, failed attempts by classic methods. Although it increases the rate of CBD cannulation, it is also an RF associated with a higher incidence of PEP [1,8,10]. Interestingly, although it has been shown that GWAC is associated with a decreased incidence of PEP when compared with the classic CAC technique, as discussed earlier, when precut sphincterotomy is performed, there is no significant difference between the two methods [23,27]. Moreover, because any failed attempt at CBD cannulation is associated with significant additional risk for PEP, early precut (after just two or three failed attempts) or even primary precut sphincterotomy (performed just before attempting CBP cannulation, reserved only for experienced endoscopists) is not associated with increased risk for PEP unlike precut sphincterotomy performed after at least five or six failed attempts [28,29,30,31]. In our study, the association of precut sphincterotomy with the incidence of PEP was not significant, with a *p*-value of 0.148.

Pancreatic stent placement for PEP prophylaxis was associated with a lower incidence of PEP in patients at high risk of developing PEP or in whom GW catheterization or contrast injection of the MPD was performed [3,10,15]. As in the case of precut sphincterotomy, the same meta-analysis performed by Tse F. et al. in 2013 demonstrates that the GWAC technique is associated with a lower incidence of PEP when pancreatic stent placement was permitted but equivalent to the classic contrast technique in studies in which no pancreatic stent placement was allowed [1]. In two meta-analyses published in 2015 and 2023 that have analyzed studies published since 1990 and included a total of more than 4350 patients, the efficacy of pancreatic stenting in terms of PEP prophylaxis was demonstrated, with an incidence in those with stent of 3.97% vs. 10.41% (OR = 0.35, 95% CI of 0.25–0.49, *p* < 0.00001) and 5.9% vs. 16.8% (OR = 0.40, 95% CI of 0.30–0.54, *p* < 0.001), respectively [32,33,34]. Similarly, in our study, pancreatic stenting was performed only after three unintentional MPD catheterizations or contrast agent injections and was associated with a decrease in PEP incidence, with an OR = 0.485, a 95% CI of 0.27–0.873, and a *p*-value = 0.016.

EPBD represents another procedural element often postulated as an RF for PEP, associated with up to 5–20% of cases [35]. Initially developed as an alternative method with no risk of bleeding or perforation, as in the case of classic sphincterotomy, an increased associated risk of PEP has been observed over time. In our endoscopy unit, the hybrid method is sometimes used, which involves a limited sphincterotomy followed by EPBD, usually after a failed attempt to remove large gallstones or when an extended sphincterotomy is not possible. Thus, in our study group, EPBD was an RF strongly associated with the incidence of PEP, with an OR = 3.258, a 95% CI of 2.011–5.28, and *p* = <0.001. Although an increased incidence of PEP is often observed in patients undergoing balloon dilation in numerous prospective randomized studies, ranging from 7.4 to 15.4%, the exact mechanism is still unknown, and the results are ultimately inconsistent or even controversial [36,37,38,39]. A randomized prospective study by Chou C.-K. et al., where patients who underwent EPBD were divided into three groups according to expansion time (<3 min, 3–5 min, and ≥5 min), showed that only an EPBD lasting under three minutes was associated with higher PEP incidence, OR = 4.942, *p* = 0.027 [39]. On the other hand, two systematic reviews have shown that, although EPBD is indeed an important RF for PEP, the mechanism is not necessarily of those initially proposed, such as papillary edema, spasm, or compression of the Wirsung duct, but rather by certain elements associated with EPBD, including longer procedure, multiple failed attempts of gallstone extraction, insufficient dilatation of the Oddi sphincter and extraction of only small or medium-sized gallstones [34,35,40,41].

Our study presents both strengths and limitations. To our knowledge, it is the first study conducted in a center in Romania that aimed to analyze the incidence of PEP and potential associated RF. Moreover, it includes a significant number of patients, treated over 5 years, with diverse biliopancreatic pathology, in a major tertiary referral endoscopy unit in Bucharest, Romania. Several strong elements, including the multivariate analysis, the comparative nature of the study, the extended period, the size of the study group, and the comparison of the obtained data with the international data, contribute to the significance of this work. On the other hand, it is a retrospective study, which brings a risk of possible missing data and incomplete selection of participants, leading to sampling bias and difficult causal relationship determination. However, our investigation can raise awareness regarding the importance of PEP as a possible complication of ERCP and can serve as a source of documentation for decision-makers in the healthcare field to further improve the endoscopy and PEP prophylaxis protocol and for a better selection of patients at high risk of developing PEP.

Although there are numerous prospective randomized controlled trials and meta-analyses available in the literature addressing the inter-relationship among RF for PEP, there is still a need for further research in this area in Romanian hospitals, as there are numerous young endoscopists still in training, which may influence the incidence rate of PEP, mainly through differences in the evolution of the learning curve, access to advanced endoscopic methods or the implementation of efficient PEP prevention strategies.

## 5. Conclusions

PEP remains the most common and serious complication of ERCP, associated with significant morbidity and occasional mortality, despite recent advances regarding the mechanisms and associated RF. Despite numerous preventive measures, both pharmacological and endoscopic, there is still no real strategy to limit the severity of PEP. Numerous RFs have been identified and proven to be involved in the etiopathogenesis of PEP, most of which are procedure-related and, therefore, modifiable. Therefore, the most important methods of PEP prevention are represented by, in addition to the methods already mentioned, a rigorous, well-established endoscopy workflow with multiple evidence-based practices implemented by experienced endoscopists. While many recommendations are widely known and easily accessible, future research is needed to further validate these results and to ensure better patient safety and improved clinical outcomes.

## Figures and Tables

**Figure 1 biomedicines-13-00727-f001:**
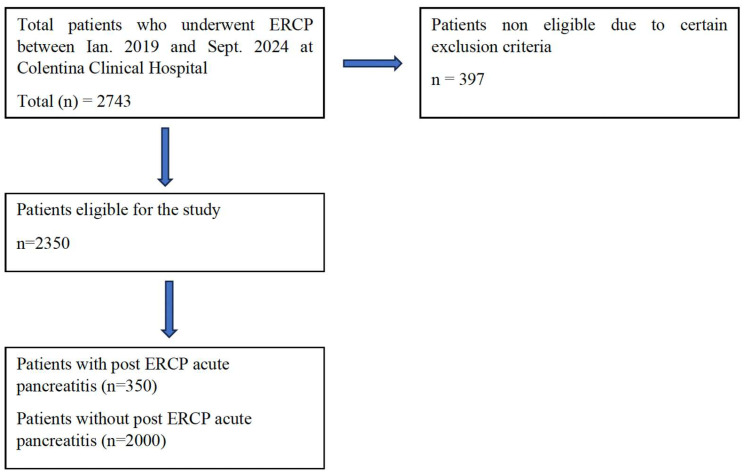
Inclusion process of the patients.

**Figure 2 biomedicines-13-00727-f002:**
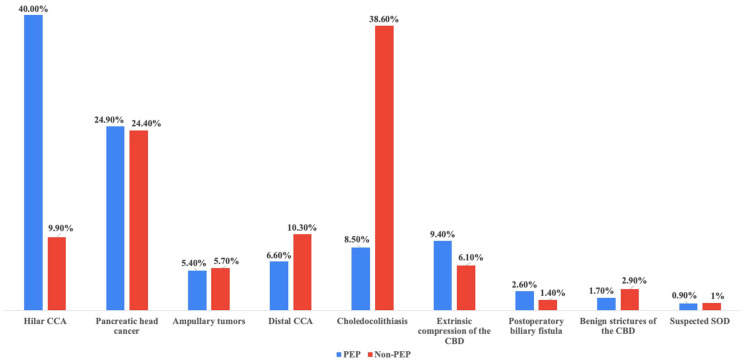
The most frequent indications for ERCP in the PEP group and in the control group. CCA—cholangiocarcinoma, SOD—Sphincter of Oddi dysfunction, CBD—common bile duct.

**Figure 3 biomedicines-13-00727-f003:**
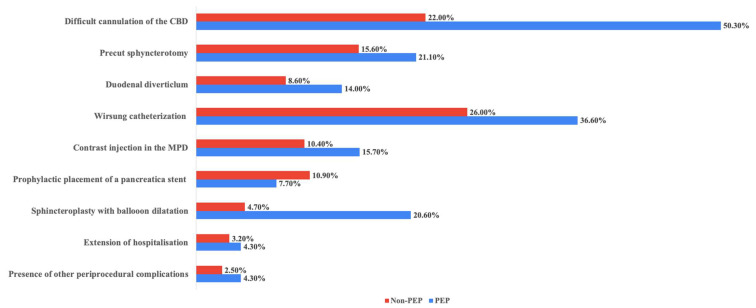
Procedural characteristics of the PEP and non-PEP group. CBD—common bile duct, MPD—main pancreatic duct.

**Figure 4 biomedicines-13-00727-f004:**
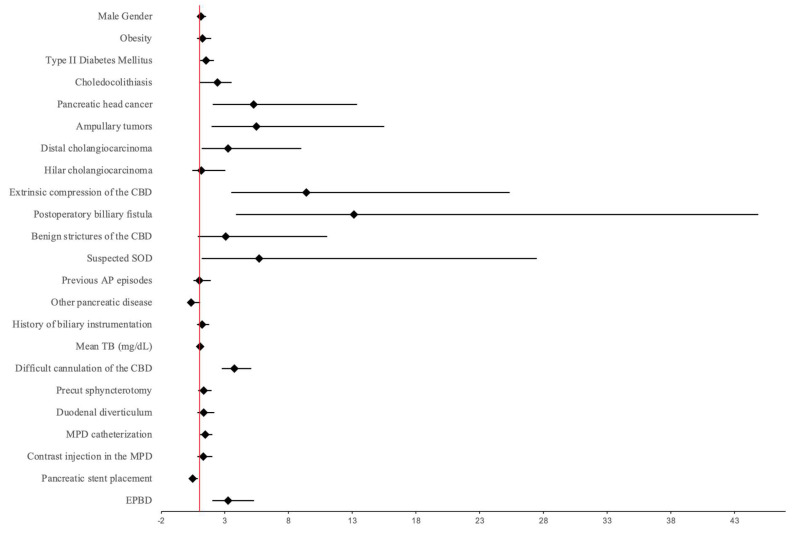
Forest plot analysis for PEP development. CBD—common bile duct, TB—total bilirubin, SOD—Sphincter of Oddi dysfunction, AP—acute pancreatitis, MPD—main pancreatic duct, EPBD—endoscopic papillary balloon dilatation.

**Table 1 biomedicines-13-00727-t001:** General characteristics of the study groups.

Variables	Non-PEP Group	PEP Group	Total
Male gender (%)	952 (47.6%)	191 (54.6%)	1111 (47.3%)
Mean age (SD)	66.25	66.5	66.29
Obesity (%)	227 (11.4%)	47 (13.4%)	274 (11.7%)
T2DM (%)	391 (19.5%)	91 (26%)	482 (20.5%)
Previous AP episodes	97 (4.9%)	19 (5.4%)	116 (4.9%)
Other pancreatic disease	100 (5%)	4 (1.1%)	104 (4.4%)
History of biliary instrumentation	309 (15.45%)	58 (16.6%)	367 (15.61%)
Mean TB (mg/dL)	9.92 mg/dL	9.23 mg/dL	9.82 mg/dL
Mortality	17 (0.85%)	8 (2.29%)	25 (1.05%)

T2DM—type 2 diabetes mellitus, SD—standard deviation, AP—acute pancreatitis, TB—total bilirubin.

**Table 2 biomedicines-13-00727-t002:** Severity classification of PEP episodes.

Variables	Values
Mean lipase value (U/L) (SD)	1955.44 (2451.78)
Severity of PEP	
Mild	337 (96.3%)
Moderate	9 (2.6%)
Severe	4 (1.1%)

SD—standard deviation, PEP—post-ERCP pancreatitis.

**Table 3 biomedicines-13-00727-t003:** The main indications for ERCP.

Indication	Non-PEP Group (n = 2000)	PEP Group (n = 350)	Total (m = 2350)
Hilar cholangiocarcinoma	197 (9.9%)	140 (40%)	337 (14.3%)
Pancreatic head cancer	488 (24.4%)	87 (24.9%)	575 (24.5%)
Ampullary tumors	113 (5.7%)	19 (5.4%)	132 (5.6%)
Distal cholangiocarcinoma	206 (10.3%)	23 (6.6%)	229 (9.7%)
Choledocholithiasis	772 (38.6%)	30 (8.5%)	802 (34.1%)
Extrinsic compression of the CBD	121 (6.1%)	33 (9.4%)	154 (6.6%)
Post-operatory biliary fistula	27 (1.4%)	9 (2.6%)	36 (1.5%)
Benign strictures of the CBD	57 (2.9%)	6 (1.7%)	63 (2.7%)
Suspected SOD	19 (1%)	3 (0.9%)	22 (0.9%)

SOD—Sphincter of Oddi dysfunction, CBD—common bile duct.

**Table 4 biomedicines-13-00727-t004:** Procedural techniques, complications, and outcome.

Variables	Total	Non-PEP (n = 2000)	PEP (n = 350)	*p*-Value
Difficult cannulation of the CBD	615 (26.2%)	440 (22%)	175 (50.3%)	<0.0001
Precut sphincterotomy	385 (16.4%)	312 (15.6%)	73 (21.1%)	0.0186
Duodenal diverticulum	220 (9.4%)	171 (8.6%)	49 (14%)	0.002
Wirsung catheterization	648 (27.6%)	520 (26%)	128 (36.6%)	<0.0001
Contrast injection in the MPD	262 (11.1%)	207 (10.4%)	55 (15.7%)	0.0043
Prophylactic placement of a pancreatic duct	244 (10.4%)	217 (10.9%)	27 (7.7%)	0.0868
Sphincteroplasty with balloon dilation	166 (7.1%)	94 (4.7%)	72 (20.6%)	<0.0001
Extension of hospitalization	78 (3.3%)	63 (3.2%)	15 (4.3%)	0.2598
Presence of other complications during the procedure	64 (2.7%)	49 (2.5%)	15 (4.3%)	0.0721
Difficult cannulation of the CBD	615 (26.2%)	440 (22%)	175 (50.3%)	<0.0001
Precut sphincterotomy	385 (16.4%)	312 (15.6%)	73 (21.1%)	0.0186

CBD—common bile duct, MPD—main pancreatic duct.

**Table 5 biomedicines-13-00727-t005:** Logistic regression analysis for PEP development.

Variables	Odds Ratio (Lower CI–Upper CI)	*p*-Value
Female Gender	1.135 (0.843–1.529)	0.404
Obesity	1.243 (0.8–1.931)	0.333
Type II Diabetes Mellitus	1.508 (1.069–2.128)	0.019
Choledocholithiasis	2.419 (1.038–3.51)	<0.001
Pancreatic head cancer	5.244 (2.058–13.362)	<0.001
Ampullary tumors	5.487 (1.94–15.516)	0.001
Distal cholangiocarcinoma	3.255 (1.178–8.992)	0.023
Hilar cholangiocarcinoma	1.144 (0.43–3.046)	0.788
Extrinsic compression of the CBD	9.401 (3.485–25.358)	<0.001
Post-operatory biliary fistula	13.139 (3.85–44.848)	<0.001
Benign strictures of the CBD	3.076 (0.858–11.032)	0.085
Suspected SOD	5.677 (1.173–27.487)	0.031
Previous AP episodes	0.989 (0.516–1.898)	0.974
Other pancreatic disease	0.363 (0.125–1.052)	0.062
History of biliary instrumentation	1.202 (0.812–1.779)	0.358
Mean TB (mg/dL)	1.049 (1.023–1.075)	<0.001
Difficult cannulation of the CBD	3.734 (2.747–5.076)	<0.001
Precut sphincterotomy	1.328 (0.904–1.95)	0.148

CBD—common bile duct, TB—total bilirubin, SOD—Sphincter of Oddi dysfunction, AP—acute pancreatitis.

## Data Availability

Data is contained within the article.

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
