# Peer review of "Incidence, Risk Factors, and Prevention Strategies for Post-ERCP Pancreatitis in Patients with Biliopancreatic Disorders and Acute Cholangitis: A Study from a Romanian Tertiary Hospital†"

_biomedicines, 2025, doi:10.3390/biomedicines13030727_

Round 1
Reviewer 1 Report
Comments and Suggestions for Authors
I congratulate the authors for the large scale study on patients undergoing ERCP. In this study, risk factors for PEP were investigated and interestingly, the findings showed that PEP developed in men and at an older age than expected. The increased incidence of PEP in those who underwent precut sphincterotomy may be due to repeated pancreatic duct cannulations prior to this procedure. So, it may be independent of the precut sphincterotomy procedure. This issue has been adequately explained in the discussion.
Author Response
We are very thankful to you for the pertinent notes; we have carefully read the comments and have revised/completed the manuscript accordingly. Our responses are given in a point-by-point manner below. All the changes to the manuscript are highlighted in yellow. We hope that, in this new form, the manuscript will be suitable for publication in the Biomedicines.
I congratulate the authors for the large scale study on patients undergoing ERCP. In this study, risk factors for PEP were investigated and interestingly, the findings showed that PEP developed in men and at an older age than expected. The increased incidence of PEP in those who underwent precut sphincterotomy may be due to repeated pancreatic duct cannulations prior to this procedure. So, it may be independent of the precut sphincterotomy procedure. This issue has been adequately explained in the discussion.
Response: Thank you for your positive feedback regarding our manuscript and for considering that the results of our study are worth publishing in Biomedicines.
Reviewer 2 Report
Comments and Suggestions for Authors
The Authors present their experience of ERCPs performed in a five -year period with particular attention to the search of risk factor for post-ERCP pancreatitis and the strategy to prevent this complication. The study includes a large amount of procedures but the results do not add any new infomations compared to previous reports published in the literature. Moreover, this is a retrospective study with a potential risk of bias (14% of patients excluded for several reasons), 15% PEP incidence higher compared to previous reports (not discussed), patients with different diseases and neoplasms, many endoscopists with presumable different expertise. The diagnosis of PEP was substantially based on clinical presentation since pancreatic lipase and radiological procedures were performed only in selected cases (there were 4 severe PEP and only 2 underwent CT examination). There are some errors in the definition of patients' subgroups (Fig. 1), the table 1 is difficult to read (there are two NON-PEP groups).
Author Response
Dear Editor-in-chief, Guest Editors, and Peer-Reviewers,
We are very thankful to you for the pertinent notes; we have carefully read the comments and have revised/completed the manuscript accordingly. Our responses are given in a point-by-point manner below. All the changes to the manuscript are highlighted in yellow. We hope that, in this new form, the manuscript will be suitable for publication in the Biomedicines.
The Authors present their experience of ERCPs performed in a five -year period with particular attention to the search of risk factor for post-ERCP pancreatitis and the strategy to prevent this complication. The study includes a large amount of procedures but the results do not add any new infomations compared to previous reports published in the literature.
Response: Thank you for your comments.
While most of the aspects highlighted in this study about the risk factors of PEP are indeed already known in the literature. However, to our best knowledge to this day, there are only two similar studies analyzing similar data, both monocentric, with a limited study group (up to 240 patients). Even if our study keeps the same format, it is important to emphasize that it stands out because of the large number of patients analyzed (more than 2000), giving legitimacy to the obtained results. In the same time, our study has identified some key findings that contradict the existing literature, such as the fact that two patient-related risk factors, young age and female sex, were not demonstrated as factors associated with the development of PEP in our study. Thus, our study provides valuable and relevant regional data, offering insights in the type of pathology and its endoscopic management in an important tertiary clinic, specific to our health care system, despite its restrospective character.
Moreover, this is a retrospective study with a potential risk of bias (14% of patients excluded for several reasons), 15% PEP incidence higher compared to previous reports (not discussed), patients with different diseases and neoplasms, many endoscopists with presumable different expertise.
Response: Thank you for your comment. We excluded patients with lack of follow-up after ERCP (patients who were transferred to other medical units immediately after the procedure) or incomplete data, especially those with incomplete endoscopic reports, with missing data such as the degree of difficulty in cannulating the CBD, pre-cut sphincterotomy or pancreatic stenting placement). Unfortunately, in 2023, a major technical problem was identified in our database containing the endoscopic reports performed in our clinic. This deficiency led to the loss or inaccessibility of a significant part of the data spanning several months, directly affecting the ability to include all patients in the study. As a result, we had to exclude a number of patients who would have been eligible for analysis.
The incidence of PEP is estimated, according to the latest literature data, between 5 and 10%. Bishay K. et al. have shown in their meta-analysis published in 2025 an approximately stady incidence since 2000, of 4.6% in all patients and 6.4% in patients without a history of previous sphincterotomy [2]. In our study, the incidence rate of PEP was 14.9%, considerably higher than those reported in the literature. Differences in incidence between different medical centers are described in the literature, and the reasons are multiple, both patient- and procedure-related. The most frequently cited are: different categories of patients treated in certain centers (some referral centers predominantly treat patients with complex bilio-pancreatic disorders, many of them with previous history of endoscopic sphincterotomy or episodes of acute pancreatitis), differences in the routine endoscopic techniques used or in the level of experience of the endoscopists who performed ERCP (the majority of prospective studies usually involve only experienced endoscopists) and, last but not least, differences in the criteria for defining acute pancreatitis and the cut-off values of serum lipase levels sampled after ERCP and considered diagnostic. In our study, we consider that there were two main factors that led to a higher incidence of PEP: the diverse pathology of the patients, a significant number of them being diagnosed with neoplastic bilio-pancreatic pathology and with a history of endoscopic sphincterotomy and, the different level of expertise of the endoscopists who performed ERCP, our clinic being also a training center for young endoscopists.
The diagnosis of PEP was substantially based on clinical presentation since pancreatic lipase and radiological procedures were performed only in selected cases (there were 4 severe PEP and only 2 underwent CT examination).
Response: Thank you for your comment. Serum lipase concetrations are a key element in the diagnosis of PEP, usually performed after the onset of symptoms suggestive of acute pancreatitis. Transient asymptomatic elevation of serum lipase or amylase occurs in up to 70% of patients after ERCP, beginning at 90 minutes and peaking approximately 4h after the procedure. Although this is well known in the endoscopic literature, there are numerous studies that have shown that a value of more than three to four times the ULN in patients without suggestive symptoms of acute pancreatitis performed two to six hours after ERCP can help to an erly diagnosis of PEP, avoid underdiagnosis. At the same time, a value lower than three to four times the ULN has a sufficiently good negative predictive value for excluding PEP (https://doi.org/10.1007/s11845-019-02089-2; https://doi.org/10.4103/jrms.JRMS_1100_17; https://doi.org/10.1111/ans.13665). Because ERCP is nowadays more commonly performed in an outpatient settings, this practice is used to make discharge decisions after the procedure. For this reason, the latest ESGE guideline on PEP prevention recommends routine determination of serum lipase two to six hours after ERCP in patients presenting with pain suggestive of acute pancreatitis or in patients to be discharged on the day of ERCP (http://dx.doi.org/10.1055/s-0034-1377875).
In our center, despite few selected cases in which serum lipase level is measured within two to six hours after ERCP, usually in patients with a very difficult ERCP, in whom the MPD was extensively cannulated or injected inadvertently, or in patients in whom the indication for ERCP was pancreatic pathology (i.e. MPD lithiasis in the setting of chronic pancreatitis), in the vast majority of patients, serum lipase levels are determined only in the presence of symptoms suggestive for acute pancreatitis. This practice allows us to avoid an over-diagnosis of PEP, which would lead to over-hospitalization, supplementary imaging without any additional clinical benefit or to over-treating which is not without risks. We also opted for performing a cross-sectional imaging techniques just in selected cases, usually severe forms, for the evaluation of complications, in accordance with international specialized guidelines (https://doi.org/10.1186/s13017-019-0247-0).
There are some errors in the definition of patients' subgroups (Fig. 1), the table 1 is difficult to read (there are two NON-PEP groups).
Response: Thank you for your comment. We have corrected these errors.
Reviewer 3 Report
Comments and Suggestions for Authors
The study examines the causes, risk factors, and prevention of post-ERCP pancreatitis (PEP). The ERCP interventions at Romania's Colentina Clinical Hospital were the subject of the retrospective, observational, single-center study.
The investigation is guided by methodical epidemiological and statistical procedures. The statistical efficacy of the study was enhanced by the utilisation of a large sample size (n=2350). Logistic regression is employed to identify PEP risk factors while accounting for confounding variables. The paper provides critical data from a tertiary hospital in Eastern Europe and compares it to the global literature.
The procedure is rigorous. In well-monitored ERCP patients, PEP was identified using the new Atlanta classification and the guidelines of the European Society of Gastrointestinal Endoscopy. Prophylactic measures that are advised include intravenous hydration and rectal diclofenac.
Detailed statistical analysis. Binary logistic regression was implemented by the authors to identify independent risk variables. The intensity and direction of the relationship are revealed by adjusted odds ratios (OR) and confidence intervals (CI).
The Western literature estimates of 2% to 10% are surpassed by the prevalence of PEP in this cohort (14.9%). This discrepancy may be attributed to the complexity of the procedure, the competence of the endoscopist, and the comorbidities of the patient.
PEP was significantly associated with a number of procedure-related risk factors (RFs), including challenging CBD cannulation (OR = 3.734, p<0.001).
Endoscopic papillary balloon dilation (OR = 3.258, p<0.001).
Catheterisation of the main pancreatic duct (OR = 1.454, p = 0.022).
Post-endoscopic retrograde cholangiopancreatography pancreatitis (PEP) was prevented by preventive pancreatic duct stents (OR = 0.485, p=0.016).
A patient-related PEP risk factor was type 2 diabetes mellitus (OR = 1.508, p=0.019). Contrary to previous research that associated younger women with increased PEP risk, female sex was not a significant risk factor (p=0.404).
Choledocholithiasis (34.1%) was the most prevalent cause of ERCP, followed by pancreatic head cancer (24.5%) and hilar cholangiocarcinoma (14.3%). The study outcomes, particularly the technical difficulty of the procedure, may have been influenced by the large proportion of oncological patients.
This investigation enhances comprehension of PEP, particularly in Romanian tertiary institutions. The results indicate that PEP is caused by procedure-related risk factors, but pancreatic duct stenting precludes it. The retrospective methodology and single-center emphasis must be taken into account when interpreting the results, despite the fact that the study was well-conducted. Additional research is required to examine the outcomes of ERCP patients and the prevention of PEP.
While the study is methodologically robust, it has a number of issues that can be adjusted:
1. Retrospective Design—Retrospective research is susceptible to selection bias and inadequate data collection. Despite the authors' diligent collection of data from medical records, it is feasible that there may be gaps in the data.
2. Monocentric Study — The findings of this Romanian tertiary hospital may not be generalisable. The expertise of the endoscopist, the procedure, and the demographics of the patient may all influence the external validity.
3. Inadequate Comparison of Direct Cannulation Techniques – The authors perform a comparison between contrast-assisted cannulation (CAC) and guidewire-assisted cannulation (GWAC), but they do not compare the two procedures. Given that GWAC is now a standard practice at their institution, it would have been advantageous to compare the results obtained prior to and following its widespread implementation.
4. Inadequate Utilisation of Prophylactic Pancreatic Stents - Notwithstanding the stent's demonstrated efficacy, only 7.7% of PEP patients were administered one. Optimisation may be necessary for the prophylactic stent implantation strategy.
5. Inadequate Routine Imaging for PEP Diagnosis—The study employed clinical symptoms and serum lipase levels to diagnose PEP. The selection of only significant cases for cross-sectional imaging (CT/MRI) may have resulted in the undervaluation of subclinical or moderate cases.
Author Response
Dear Editor-in-chief, Guest Editors, and Peer-Reviewers,
We are very thankful to you and for the pertinent notes; we have carefully read the comments and have revised/completed the manuscript accordingly. Our responses are given in a point-by-point manner below. All the changes to the manuscript are highlighted in yellow. We hope that, in this new form, the manuscript will be suitable for publication in the Biomedicines.
The study examines the causes, risk factors, and prevention of post-ERCP pancreatitis (PEP). The ERCP interventions at Romania's Colentina Clinical Hospital were the subject of the retrospective, observational, single-center study. The investigation is guided by methodical epidemiological and statistical procedures. The statistical efficacy of the study was enhanced by the utilisation of a large sample size (n=2350). Logistic regression is employed to identify PEP risk factors while accounting for confounding variables. The paper provides critical data from a tertiary hospital in Eastern Europe and compares it to the global literature. The procedure is rigorous. In well-monitored ERCP patients, PEP was identified using the new Atlanta classification and the guidelines of the European Society of Gastrointestinal Endoscopy. Prophylactic measures that are advised include intravenous hydration and rectal diclofenac. Detailed statistical analysis. Binary logistic regression was implemented by the authors to identify independent risk variables. The intensity and direction of the relationship are revealed by adjusted odds ratios (OR) and confidence intervals (CI).
Response: Thank you for your positive feedback regarding our manuscript.
The Western literature estimates of 2% to 10% are surpassed by the prevalence of PEP in this cohort (14.9%). This discrepancy may be attributed to the complexity of the procedure, the competence of the endoscopist, and the comorbidities of the patient. PEP was significantly associated with a number of procedure-related risk factors (RFs), including challenging CBD cannulation (OR = 3.734, p<0.001). Endoscopic papillary balloon dilation (OR = 3.258, p<0.001). Catheterisation of the main pancreatic duct (OR = 1.454, p = 0.022). Post-endoscopic retrograde cholangiopancreatography pancreatitis (PEP) was prevented by preventive pancreatic duct stents (OR = 0.485, p=0.016).
Response: Thank you for your comment. We have given explanations regarding the elevated PEP prevalence in our assessment.
Serum lipase concetrations are a key element in the diagnosis of PEP, usually performed after the onset of symptoms suggestive of acute pancreatitis. Transient asymptomatic elevation of serum lipase or amylase occurs in up to 70% of patients after ERCP, beginning at 90 minutes and peaking approximately 4h after the procedure. Although this is well known in the endoscopic literature, there are numerous studies that have shown that a value of more than three to four times the ULN in patients without suggestive symptoms of acute pancreatitis performed two to six hours after ERCP can help to an erly diagnosis of PEP, avoid underdiagnosis. At the same time, a value lower than three to four times the ULN has a sufficiently good negative predictive value for excluding PEP (https://doi.org/10.1007/s11845-019-02089-2; https://doi.org/10.4103/jrms.JRMS_1100_17; https://doi.org/10.1111/ans.13665). Because ERCP is nowadays more commonly performed in an outpatient settings, this practice is used to make discharge decisions after the procedure. For this reason, the latest ESGE guideline on PEP prevention recommends routine determination of serum lipase two to six hours after ERCP in patients presenting with pain suggestive of acute pancreatitis or in patients to be discharged on the day of ERCP (http://dx.doi.org/10.1055/s-0034-1377875).
In our center, despite few selected cases in which serum lipase level is measured within two to six hours after ERCP, usually in patients with a very difficult ERCP, in whom the MPD was extensively cannulated or injected inadvertently, or in patients in whom the indication for ERCP was pancreatic pathology (i.e. MPD lithiasis in the setting of chronic pancreatitis), in the vast majority of patients, serum lipase levels are determined only in the presence of symptoms suggestive for acute pancreatitis. This practice allows us to avoid an over-diagnosis of PEP, which would lead to over-hospitalization, supplementary imaging without any additional clinical benefit or to over-treating which is not without risks. We also opted for performing a cross-sectional imaging techniques just in selected cases, usually severe forms, for the evaluation of complications, in accordance with international specialized guidelines (https://doi.org/10.1186/s13017-019-0247-0).
A patient-related PEP risk factor was type 2 diabetes mellitus (OR = 1.508, p=0.019). Contrary to previous research that associated younger women with increased PEP risk, female sex was not a significant risk factor (p=0.404).
Response: Thank you for your comment.
Female sex, personal history of acute pancreatitis and age below 55 years old are in fact well-known risk factors associated with PEP. In our study, the only patient-related risk factor associated with PEP was type 2 diabetes mellitus. Possible resons for this may be: the predominance of male gender in our study group, both in the total group of patients (66.29%) and in the group of patients who developed PEP (66.5%), the predominance of neoplastic biliary pathology, more present in men and more common as an indication of ERCP in our study, and the retrospective nature of the study, which may lead to some patient selection bias. At the same time, the fact that the female sex is not a demonstrated risk fator can be considered an original finding of this study, but for which further quality reasearch in this area is obviously needed. Moreover, there are other studies, some of them prospective or even randomized, which have not identified female sex as a risk factor associated with PEP, such as those conducted by Christensen et al. 2004 (https://doi.org/10.1016/S0016-5107(04)02169-8), Meister et al. 2011 (http://dx.doi.org/10.1055/s-0030-1256194) or Andriulli et al. 2004 (https://doi.org/10.1016/S1542-3565(04)00295-2).
Choledocholithiasis (34.1%) was the most prevalent cause of ERCP, followed by pancreatic head cancer (24.5%) and hilar cholangiocarcinoma (14.3%). The study outcomes, particularly the technical difficulty of the procedure, may have been influenced by the large proportion of oncological patients.
Response: Thank you for your comment. Indeed, in our study group, malignant indications for ERCP, such as pancreatic head cancer and hilar or distal cholangiocarcinoma, wer predominant, thus contributing to to an increased procedural complexity due to obvious reasons. However, we consider this results as relevant, since they reflect real-world data ncountered in a tertiary referral hospital, moreover, basic endoscopic techniques or prophylactic measures were equally applied regardless of ERCP indication. Future studies that can evaluate the relationship betwee etiology and the level of technical difficulty can brin valuable data regarding this subject.
This investigation enhances comprehension of PEP, particularly in Romanian tertiary institutions. The results indicate that PEP is caused by procedure-related risk factors, but pancreatic duct stenting precludes it. The retrospective methodology and single-center emphasis must be taken into account when interpreting the results, despite the fact that the study was well-conducted. Additional research is required to examine the outcomes of ERCP patients and the prevention of PEP.
Response: Thank you for your positive feedback regarding our manuscript and for stressing out the strengths of our research.
While the study is methodologically robust, it has a number of issues that can be adjusted:
1. Retrospective Design—Retrospective research is susceptible to selection bias and inadequate data collection. Despite the authors' diligent collection of data from medical records, it is feasible that there may be gaps in the data.
Response: Thank you for your comment. We have acknowledged this limitation of our study.
Monocentric Study — The findings of this Romanian tertiary hospital may not be generalisable. The expertise of the endoscopist, the procedure, and the demographics of the patient may all influence the external validity.
Response: Thank you for your comment.
We recognise the monocentric caracter of our study and the drawbacks that comes with it. Endoscopist’s experience, patient demographics si caracterul de ocasional training centre can impact generalizability or external validity but, on the other hand, our study provides real world data and insights into clinical practice in Romania from a relevant referral endoscopy clinic, and contributes to the understanding of PEp ethio-pathogenesis at regional level. Although further multicenter studies would bring valuable data and would validate the results of this study, our findings are similar to previous studies, which further suggest the generalizability of our observations.
Inadequate Comparison of Direct Cannulation Techniques – The authors perform a comparison between contrast-assisted cannulation (CAC) and guidewire-assisted cannulation (GWAC), but they do not compare the two procedures. Given that GWAC is now a standard practice at their institution, it would have been advantageous to compare the results obtained prior to and following its widespread implementation.
Response: Thank you for your comment. Although traditionally, the most widely used method of CBD cannulation has been the CAC technique, numerous prospective studies and meta-analyses have demonstrated the superiority of the GWC technique method in terms of cannulation rate and incidence of PEP. For this reason, it is currently the preferred method of cannulation, including our centre. However, my access to retrospective data is limited to the hospital's current computer system, implemented in January 2019. Although the research idea is good and would provide valuable information, patient data prior to this date are available only in separate physical or electronic archives, which are unfortunately difficult to access and require additional data retrieval procedures, which would make comparison of the two techniques in our batch of patients impossible.
Inadequate Utilisation of Prophylactic Pancreatic Stents - Notwithstanding the stent's demonstrated efficacy, only 7.7% of PEP patients were administered one. Optimisation may be necessary for the prophylactic stent implantation strategy.
Response: Thank you for your comment.
While we acknowledge the usefulness and the suboptimal utilization of prophylactic main pancreatic duct stent placement, it’s use was limited, according to the clinic protocol, to selected high-risk cases, particularly in patients where the main pancreatic duct was cannulated at least 3 times or injected with contrast agent. Factors such as procedural complexity or endoscopist expertise may have influenced this decision. We will certainly take the results of this study into consideration to better adjust this protocol and to enhance the utilization of this preventive measure in our center.
Inadequate Routine Imaging for PEP Diagnosis—The study employed clinical symptoms and serum lipase levels to diagnose PEP. The selection of only significant cases for cross-sectional imaging (CT/MRI) may have resulted in the undervaluation of subclinical or moderate cases.
Response: Thank you for your comment.
Serum lipase concetrations are a key element in the diagnosis of PEP, usually performed after the onset of symptoms suggestive of acute pancreatitis. Transient asymptomatic elevation of serum lipase or amylase occurs in up to 70% of patients after ERCP, beginning at 90 minutes and peaking approximately 4h after the procedure. Although this is well known in the endoscopic literature, there are numerous studies that have shown that a value of more than three to four times the ULN in patients without suggestive symptoms of acute pancreatitis performed two to six hours after ERCP can help to an erly diagnosis of PEP, avoid underdiagnosis. At the same time, a value lower than three to four times the ULN has a sufficiently good negative predictive value for excluding PEP (https://doi.org/10.1007/s11845-019-02089-2; https://doi.org/10.4103/jrms.JRMS_1100_17; https://doi.org/10.1111/ans.13665). Because ERCP is nowadays more commonly performed in an outpatient settings, this practice is used to make discharge decisions after the procedure. For this reason, the latest ESGE guideline on PEP prevention recommends routine determination of serum lipase two to six hours after ERCP in patients presenting with pain suggestive of acute pancreatitis or in patients to be discharged on the day of ERCP (http://dx.doi.org/10.1055/s-0034-1377875).
In our center, despite few selected cases in which serum lipase level is measured within two to six hours after ERCP, usually in patients with a very difficult ERCP, in whom the MPD was extensively cannulated or injected inadvertently, or in patients in whom the indication for ERCP was pancreatic pathology (i.e. MPD lithiasis in the setting of chronic pancreatitis), in the vast majority of patients, serum lipase levels are determined only in the presence of symptoms suggestive for acute pancreatitis. This practice allows us to avoid an over-diagnosis of PEP, which would lead to over-hospitalization, supplementary imaging without any additional clinical benefit or to over-treating which is not without risks. We also opted for performing a cross-sectional imaging techniques just in selected cases, usually severe forms, for the evaluation of complications, in accordance with international specialized guidelines (https://doi.org/10.1186/s13017-019-0247-0).
Round 2
Reviewer 2 Report
Comments and Suggestions for Authors
The paper has been revised according to my previous suggestions. I have no further comments.
Reviewer 3 Report
Comments and Suggestions for Authors
Thank you for the modifications that you made. The article is now suitable for publication.